METHODS

# A multiscale computational framework for the development of spines in molluscan shells

**Derek E. Moulton**[1]*, **Nathanaël Aubert-Kato**[2], **Axel A. Almet**[3,4], **Atsuko Sato**[5]

**1** Mathematical Institute, University of Oxford, Oxford, United Kingdom, **2** Department of Information Sciences, Ochanomizu University, Tokyo, Japan, **3** NSF-Simons Center for Multiscale Cell Fate Research, University of California, Irvine, California, United States of America, **4** Department of Mathematics, University of California, Irvine, California, United States of America, **5** Department of Biology, Ochanomizu University, Tokyo, Japan

* moulton@maths.ox.ac.uk

**Data Availability Statement:** The version of the algorithm used in this article as well as exploration results and scripts used to generate the relevant figures are available on Zenodo at link: http://doi.

## Abstract

From mathematical models of growth to computer simulations of pigmentation, the study of shell formation has given rise to an abundant number of models, working at various scales. Yet, attempts to combine those models have remained sparse, due to the challenge of combining categorically different approaches. In this paper, we propose a framework to streamline the process of combining the molecular and tissue scales of shell formation. We choose these levels as a proxy to link the genotype level, which is better described by molecular models, and the phenotype level, which is better described by tissue-level mechanics. We also show how to connect observations on shell populations to the approach, resulting in collections of molecular parameters that may be associated with different populations of real shell specimens. The approach is as follows: we use a Quality-Diversity algorithm, a type of black-box optimization algorithm, to explore the range of concentration profiles emerging as solutions of a molecular model, and that define growth patterns for the mechanical model. At the same time, the mechanical model is simulated over a wide range of growth patterns, resulting in a variety of spine shapes. While time-consuming, these steps only need to be performed once and then function as look-up tables. Actual pictures of shell spines can then be matched against the list of existing spine shapes, yielding a potential growth pattern which, in turn, gives us matching molecular parameters. The framework is modular, such that models can be easily swapped without changing the overall working of the method. As a demonstration of the approach, we solve specific molecular and mechanical models, adapted from available theoretical studies on molluscan shells, and apply the multiscale framework to evaluate the characteristics of spines from three distinct populations of *Turbo sazae*.

## Author summary

Connecting genotype to phenotype is a fundamental goal in developmental biology. While many studies examine this link in model organisms for which gene regulatory networks are well known, for non-model organisms, different techniques are required, and

org/10.5281/zenodo.7762325 Sets of (g_1, g_2) parameters for each population was integrated in the code repository https://bitbucket.org/AubertKato/sazae_evo.

**Funding:** This work was supported by the Royal Society International Exchanges Cost Share, IEC \R3\193090, which enabled DEM to visit AS and NAK, and by JSPS Bilateral Program, JSPSBP120205703, which enabled AS to visit DEM, and by JSPS KAKENHI, JP21H04434, which helped fund NAK. The funders had no role in study design, data collection and analysis, decision to publish, or preparation of the manuscript.

**Competing interests:** The authors have declared that no competing interests exist.

multiscale computational modeling offers a promising direction. In this paper, we develop a framework linking molecular-scale interactions to tissue-level growth and mechanics to organ-level characteristics in order to investigate spine formation in *T. sazae*, a species of mollusc that displays remarkable phenotypic plasticity in spine form. Our analysis uncovers a subtle but statistically significant difference in spine form between shell specimens collected from three different localities in Japan. Moreover, by tracing the difference in form through parametric differences in the multiscale framework, we provide mechanistic insight as to how environmental differences may translate to a change in form. The methodology we present may readily be extended to more detailed modeling of this system, and the conceptual framework is amenable for multiscale analysis in other systems.

This is a *PLOS Computational Biology* Methods paper.

## 1 Introduction

One of the great challenges in computational biology is linking information across scales [1]. While a number of advances have been made in recent decades, and increasingly sophisticated and specialised computational software packages have been developed for particular scales, there remain numerous fundamental hurdles in reaching a general robust framework for multiscale modeling in biology. For instance, the past 20 years has seen huge advancements in the field of tissue biomechanics, both in terms of adapting the mathematics of classical continuum mechanics to a biological setting, and in terms of improving computational and analytical tool kits [2]. Similar improvements have been made in the field of computational cell biology [3, 4] The cell and tissue scales are only separated by one level, and are strongly intertwined, and with causality in both directions, e.g., tissue growth is understood as an averaging of cell division, while forces at the tissue level impact cell signaling. Nevertheless, connecting models across even these two 'neighboring' scales very much remains an open challenge [5].

In this paper, we consider pattern formation from the point of view of three distinct scales: the molecular scale, the tissue scale, and the organ scale. Most patterns that can be discerned at the organ/organism scale are products of chemical activity within and between cells, and tissue-level growth and mechanics. But linking these scales is a daunting challenge. The molecular and tissue scales not only present distinct computational domains, but with different governing physics (mass action law versus continuum mechanics, e.g.) such that the computational variables in one scale do not clearly map to those in the other. While multi-scale computational toolboxes have been developed in some areas, e.g. the heart, multi-scale modeling in biology remains one of the great challenges [1, 2]. Our objective here is to develop a system of analysis with the potential to make quantitative predictions on form that can be directly compared with physical specimens, and in a manner that enables for qualitative observations on the mechanisms of pattern formation. To this end, we consider as a particular system the formation of spines in molluscan shells.

In the context of growth and organ/organism-level pattern formation, molluscan shells provide a unique case study. Mollusca are the second largest phylum in the animal kingdom, displaying a huge diversity in form [6]. The development of the mollusc can largely be understood by examining its shell, as the shell provides a well-preserved spatiotemporal record

of the development of the animal. In particular, the simplicity of the shell secretion process, which occurs iteratively at the shell opening by a tissue called mantle, makes shells an attractive candidate for multiscale modeling of pattern formation. Numerous studies have developed mathematical descriptions of shell geometry (as reviewed, for example, in [7, 8]). Such approaches trace as far back as 1838 [9], and have flourished since the pioneering work of Raup [10] in launching the field of theoretical morphology. However, the parameters that are used to simulate form in seashells are typically disconnected from the biological processes that underlie the shell form. Even in the case of geometric models that are constructed in such a way so as to link to the biological process of growth [11–13], nevertheless, there remains a significant gap between the mathematics used to represent the shell form and any biological processes at the cell or sub-cellular scale occurring during shell secretion.

Our interest in this work is the formation of spines. Spines are a particularly prominent feature on a number of mollusc shells, and form a canonical example of convergent evolution [14]. The formation of spines was shown to have a clear mechanical underpinning [15] through tissue-level modeling of mantle and shell growth. An important ingredient in [15] was heterogeneity: a 'pre-pattern', most likely biochemical, is needed in the growth or mechanical properties as input to the mechanical model. The question then remains: what processes at the lower scales are driving such a pre-pattern? This question highlights the need to connect tissue mechanics to biochemistry. A number of biochemical models of shells also exist, aimed at reproducing pigmentation patterns on shells through reaction-diffusion or neural-based modeling [16–18]. However, to our knowledge, such biochemical pigmentation pattern formation has never been connected to tissue-level structural pattern formation. Spines thus form an ideal case study for multiscale pattern formation, as they provide a distinctive, mostly two-dimensional pattern that shows intraspecific morphological variation and that seems to have a clear link between the molecular and tissue scales, though one whose details are a mystery. To formulate a mathematical description that explicitly connects these disparate conceptual areas is one of the objectives of this study.

In particular, in this work we are motivated by an extraordinary example of phenotypic plasticity displayed by the species *Turbo sazae* (Fukuda 2017) (formally confused with *Turbo cornutus* [19]), a marine gastropod found in South Korea and Japan. This species often displays prominent spines, or horns as they are often referred (the species is colloquially known as the 'horned turban shell' and generally called 'Sazae' in Japan)—we use the terms interchangeably in this study. However, a wide variety of the size, pattern, and even presence, of spines has been observed within the species [20]. Previous studies suggested that the plasticity found in spine form is strongly linked to the environment. For example, ecological studies showed that populations from an exposed shore grow large horns, whereas populations from a protected inner bay display only a short horn or no horn at all [21]. Moreover, horned specimens that were transferred to a pool without strong waves stopped growing horns [22]. Other studies have suggested the influence of the genotype rather than that of the environment [23]. However, a recent population genetics study showed that, even in populations of similar genetic background, there was a large phenotypic difference in a single bay in Akita prefecture depending on the location inside the bay [24]. This remarkable diversity of phenotype within a species provides the backdrop to study multiscale pattern formation.

In order to investigate whether environmental conditions may have any discernible impact on spine form, we perform a morphometric analysis of spine shape from actual shells of *T. sazae* obtained from three different locations in the Kanto area in Japan, each with distinct, qualitative environmental characteristics. The three locations, noted in Fig 1(a), are Najima in Hayama (here we call 'Hayama'), Jyogashima in Miura ('Jyogashima'), Kanagawa Prefecture, and Okinoshima in Tateyama ('Tateyama'), Chiba prefecture. Representative sample shells

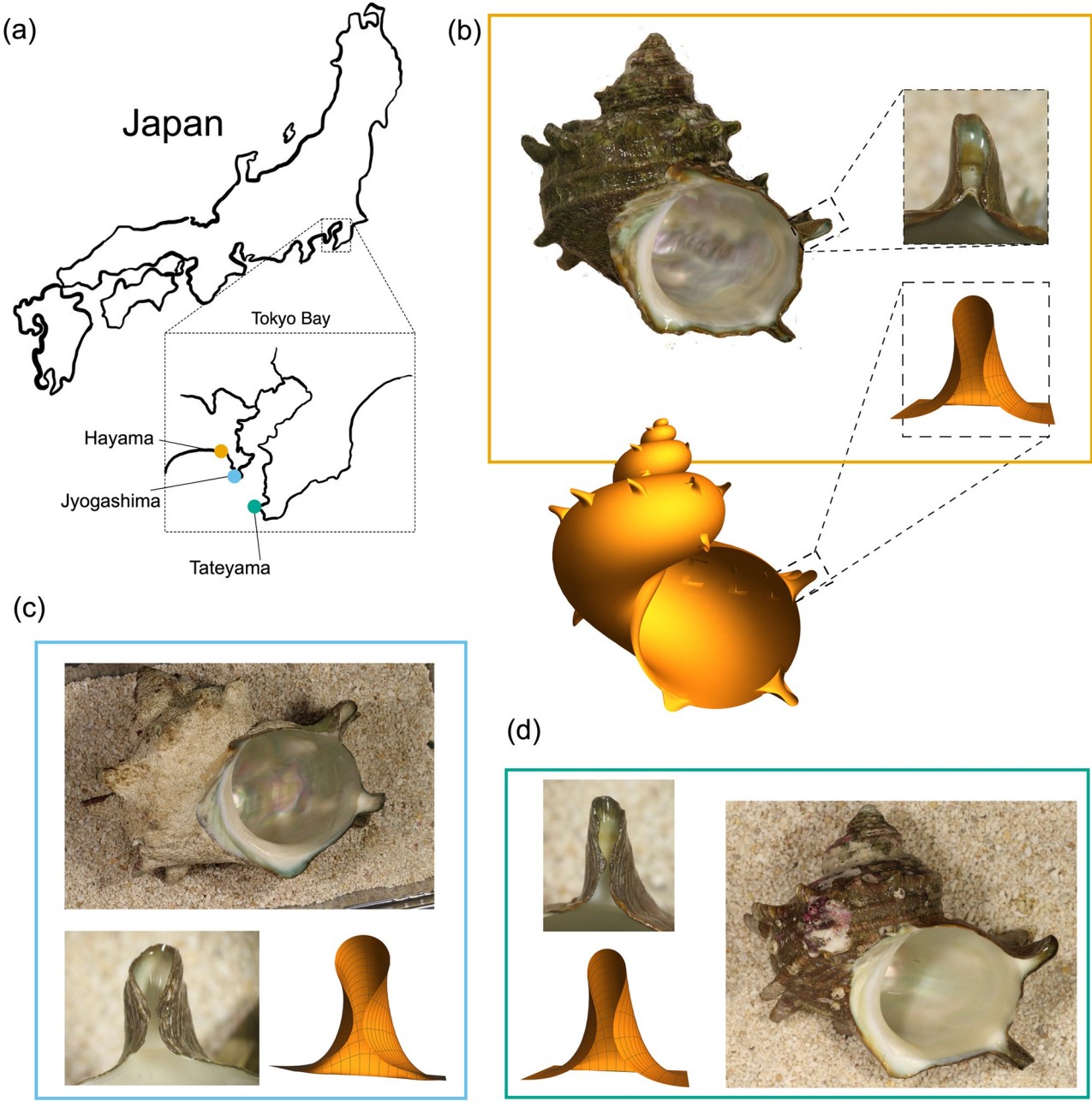

**Fig 1.** (a) A map of Japan with zoom-in on Tokyo bay, showing the locations of the three populations from which we have obtained *T. sazae* shells. Sample shells from each of the three populations: Hayama (b), Jyogashima (c), and Tateyama (d), including close-up image of a spine, and mathematical reconstructions of spines from our modeling framework (see S1 Text Section 7). Maps adapted from mapchart.net.

from each of the populations are shown in Fig 1(b)–1(d), as well as simulated spines. For visualisation purposes, we also include a simulated shell with spines superimposed. At a glance, shells from the different locations appear very similar. Another objective in our analysis is therefore to determine whether any morphological differences exist between spines from the different populations, and if so, to both quantify and understand these differences.

While morphological differences have been observed and quantified in relation to environmental characteristics in molluscs in previous studies, for example reviewed in [25], a key difference in the present study is that we seek a *mechanistic* understanding: what change(s) in the molecular process may account for observed morphological differences? To do this, we construct a mathematical framework for spine formation that enables us to probe the characteristics of spines in *T. sazae* within a multiscale setting, specifically linking biochemical pattern formation with tissue-level growth and mechanics. In order to link the mathematical framework to spine form, from each shell we extract quantitative characteristics of the shape, and then seek the input parameters to the model that best produce the given shape as output. Connecting molecular-scale interactions to population-level phenotypic differences fundamentally spans all the scales, and a full description is a monumental task. This is particularly true when considering that *T. sazae* is a non-model organism, for which almost no genetic studies have been done and no genetic manipulation techniques are available. Probing the link between genotype and phenotype for such organisms may require new techniques and approaches, and computational modeling has the potential to both shed light on developmental mechanisms and also open doors to new biological studies.

Given these fundamental challenges, in this first instance we will opt for modeling simplicity wherever feasible, focusing more on conceptual methodology than detailed biology. As such, the results we present are intrinsically linked to the underlying assumptions and simplicity of the models, and firm conclusions on *T. sazae* will not be possible. On the other hand, a key benefit of our approach is the 'plug and play' aspect of the modeling. More complex models may naturally be incorporated, and the procedure we present for linking the models and analyzing their success at generating shapes that match real spines would be unchanged. We return to this point in the Discussion section.

This paper is organized as follows. In Section 2, we outline the modeling framework and methods for shell collection and spine data extraction, and how we approach the flow of information across scales. In Section 3, we illustrate the range of output spine shapes produced by the model, and report the results of data extraction and model parameter fitting. Here, we identify subtle differences in spine shape between the populations and examine the corresponding difference in parameters at the biochemical scale. In this way, we can speculate as to which molecular interactions might be influenced by the phenotypic switch, such as differing environmental pressures for each population. A summary and discussion of modeling and experimental extensions needed to generate more biologically grounded conclusions is given in Section 4.

## 2 Methods

### 2.1 Spine formation in molluscs

Molluscan shells form via an accretion process that occurs at the shell edge by a thin elastic organ called the mantle. The mantle lines the inner surface of the shell, such that during the incremental growth of a new layer of shell material, the mantle extends slightly beyond the already calcified edge of the shell, adheres to the shell edge, and secretes proteins which subsequently harden into a new layer of shell. This is shown schematically in Fig 2(b).

Some of the most interesting features on molluscan shells are the so-called ornamentations, three-dimensional structural patterns such as ribs, spines, and tubercles that appear as deviations from the simple logarithmic shell coiling [20]. These organism-level patterns are diverse, occasionally quite complex (e.g., fractal-like spines on certain species of Muricidea), and tend to be quite regular within a specimen. A suite of models have been formulated to understand different types or aspects of shell ornamentation in terms of the physical forces involved in the shell growth process, by considering the mechanical interaction of the shell-secreting soft mantle and

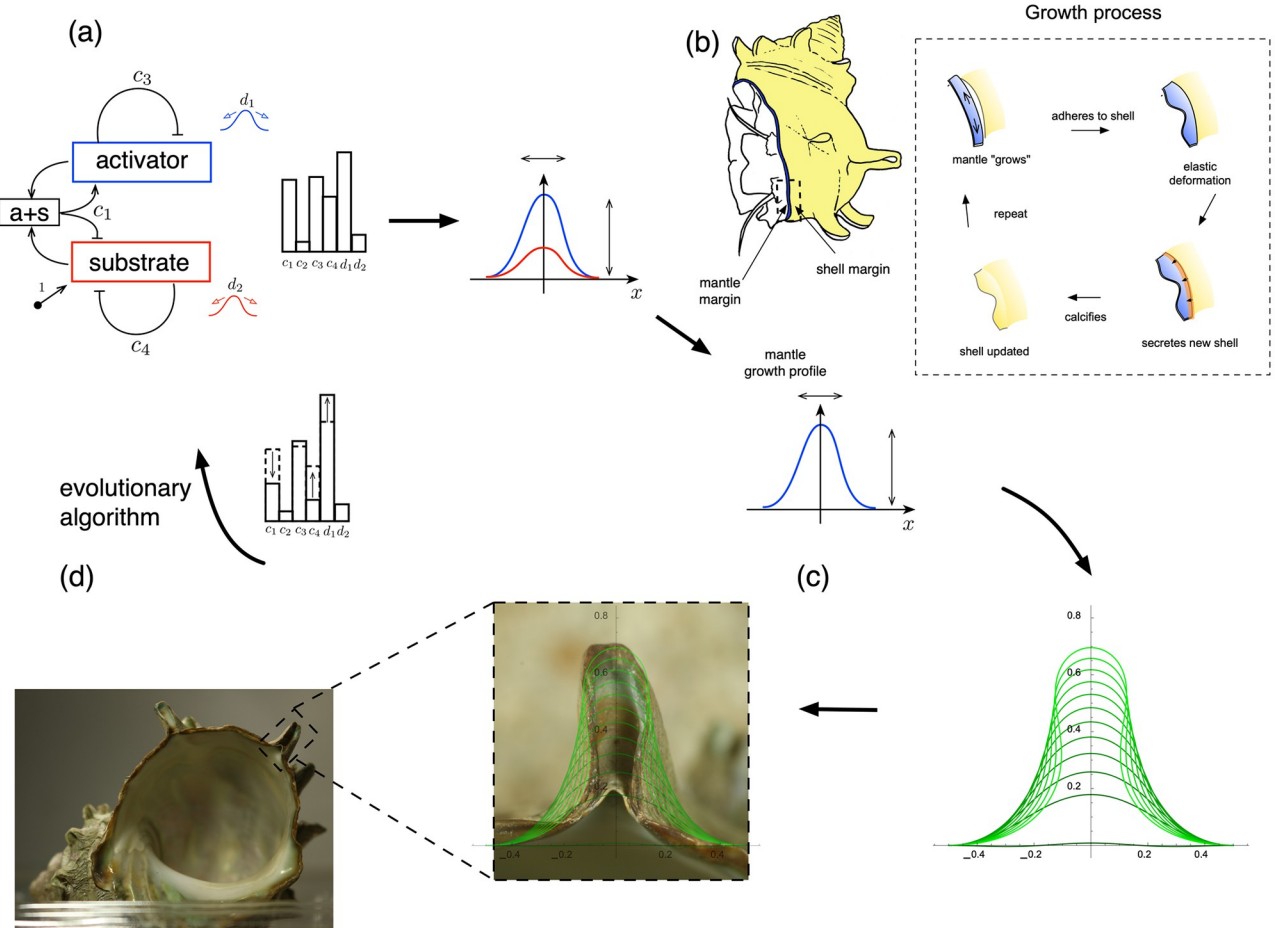

**Fig 2. Modeling framework schematic.** A biochemical system, modeling an Activator-Substrate reaction (a), potentially creates a spatial pattern, depending on the choice of six parameters. This biochemical pattern is passed to the tissue scale (b) and defines the rate and width of mantle growth and secretion. From this mechanical process, a spine shape emerges, which is then compared with an image of an actual spine (c). For a given shell, we then seek the biochemical parameters such that the model output best matches the real spine shape, (d).

the rigid shell aperture to which it attaches during shell formation [15, 26–31]. The key conceptual idea is that if the secreting mantle is longer than the shell edge, then attachment to the shell may induce deformation of the mantle, thereby influencing the shape of the subsequent layer. In other words, the shell edge determines the shape of the mantle, while the shape of the mantle determines the evolving shape of the shell edge. Previous mechanics-based models have been successful in providing a physical basis for a number of previously unexplained features on shells, including the interlocking of bivalve shells [32] and the strange meandering shape of certain ammonites [33]. However, while such models can be informative as to the basic mechanisms of pattern formation, e.g., a mismatch between mantle growth and shell secretion, they are limited by construction at the continuum tissue scale. Thus, the question remains of how the appropriate ingredients underlying the model input/assumptions are generated.

## 2.2 Model

We restrict our attention to the formation of a single spine, described at the tissue-level by the mechanics of growing elastic rods, an extension of a previous model [15], and with

heterogeneity (pre-pattern) in growth explicitly emerging as the output of an underlying biochemical model borrowed from a previous work on shell pigmentation [17]. Using output from the biochemical model as input to the tissue mechanics model to simulate a spine is novel but nevertheless straightforward; the major challenge is in (1) doing so in such a way that the resultant shapes have features corresponding to those observed in the collected shells, and (2) searching the model parameter space for values that best match the extracted spine data.

The premise of our analysis is that spine formation results from a spatially inhomogeneous shell growth process, with inhomogeneity at the level of mantle deformation and shell secretion deriving from underlying inhomogeneity at the cellular/subcellular scales due to biochemical processes. Our methodology and modelling approach is illustrated schematically in Fig 2, and consists of four components: biochemical pattern formation, tissue-level spine formation via growth and elastic deformation, extraction of growth parameters to match spine shapes from shell specimens, and corresponding biochemical parameter determination. The broad idea is that we pose a set of biochemical equations, whose behaviour is governed by a small number of parameters, $\mathcal{S} := \{c_1, c_2, c_3, c_4, d_1, d_2\}$. Spatial pattern formation at the molecular level is then passed to the tissue-level, where the same form of spatial pattern is input as an inhomogeneous profile for a mechanical model of the growth and spine secretion. The output at that level is the shape of a spine across a time sequence. This output can then be compared with images of actual spines, such that for a given spine from a real shell, we seek the parameters $\mathcal{S}$ for which the shape output is the best match. Repeating this process for a large number of spines from different populations, we can then aim to quantify any population-level differences across the scales.

Each of these methodological components is outlined further below.

**2.2.1 Molecular-level: Biochemical pattern formation.** Previous works on the mechanics of spine formation have shown that generating realistic spine shapes requires the presence of spatial inhomogeneity in the growth/secretion field [15]. To provide a molecular basis for this inhomogeneity, we hypothesize the existence of two or more chemical species interacting in such a way as to generate a spatially inhomogeneous biochemical pattern. This chemical interaction would ideally be derived from an appropriate gene regulatory network for spine formation; however, *T. sazae* (and indeed all spine-forming molluscs), are not model organisms, so no protein-protein interaction network nor gene network is available. Therefore, for simplicity in this first instance, we take a more abstract and phenomenological approach, employing a two component reaction-diffusion equation. In particular, we adopt the Activator-Substrate model of Fowler et al. [17], in which the concentrations of an activator, $a(x, t)$, and substrate, $s(x, t)$, satisfy the following equations:

$$\frac{\partial a}{\partial t} = \frac{sa^2}{1 + c_1 a^2} + c_2 s - c_3 a + d_1 \frac{\partial^2 a}{\partial x^2}$$

$$\frac{\partial s}{\partial t} = 1 - \frac{sa^2}{1 + c_1 a^2} - c_4 s + d_2 \frac{\partial^2 s}{\partial x^2}.$$

(1)

These equations model an autocatalytic process, shown schematically in Fig 2(a). In these non-dimensionalised equations (see S1 Text Section 1 for the dimensional equations and scalings), the behaviour and potential for pattern formation is governed by six parameters, $\mathcal{S} := \{c_1, c_2, c_3, c_4, d_1, d_2\}$. In this model, the activator and substrate diffuse along the single spatial direction, $x$, with diffusion constants, $d_1$, and $d_2$, respectively. The activator decays with rate $c_3$, is produced directly from the substrate at rate $c_2$, while the first term on the right-hand side of the top equation models the autocatalytic process, which requires the presence of

substrate and whose rate saturates with increasing activator at a value depending on $c_1$. The substrate decays at rate $c_4$, is produced at constant rate equal to one in non-dimensional form, and is consumed by the autocatalytic process.

This model was shown in Fowler et al. [17] to generate stable patterns for certain parameter regimes, starting from a spatially uniform initial state, given a small degree of random noise that drives the system away from an unstable initial homogeneous state. This particular model has the advantages of being simple to solve with a low-dimensional parameter space and moreover with stable spatial profiles characterised by a smooth and relatively simple form. These appealing characteristics make the model a suitable candidate for our idealised multiscale modeling framework. Nevertheless, we must acknowledge that we have no physical rationale to use (1); the choice is made out of simplicity and due to a lack of knowledge of a more realistic model. A related downside of this modeling choice is that it is very difficult to connect the input parameters to specific molecular concentrations that could, in principle, be measured in future experiments. However, we note that our objective is largely to demonstrate a methodology. The basic ideas we outline are equally amenable to more complex molecular modeling (e.g., gene regulatory network), though of course a more complex molecular model would introduce other difficulties, such as a higher dimensional parameter space or more complex patterns as output that are less straightforward to pass to the higher scale.

**2.2.2 Tissue-level: Spine formation.**   At the tissue-level, we follow the basic methodology of Chirat et al. [15], under which spines emerge as a predictable outcome of the mechanics of the mantle growth and shell secretion process. During shell secretion, the mantle margin attaches to the rigid shell aperture and secretes a new layer of shell material. As shown schematically in Fig 2(b), it is through this physical interaction that the potential for spines (or other shell ornamentation) appears: if the secreting mantle edge is longer than the current aperture, a deformation of the soft tissue will be generated, i.e., a mechanical buckling of the mantle will be induced by attachment to the shorter shell edge. As the mantle subsequently secretes new shell material, the deformed shape of the mantle edge becomes calcified in the new layer of shell. As the mantle continues to grow and secrete, the buckled shape evolves, with the final spine shape controlled by the length mismatch and mechanical properties. The approach at the tissue level is to model the physical interaction between the soft, growing mantle and the rigid shell to which it adheres.

Mathematically, we treat the mantle edge as a growing planar elastic rod, described by the parameterised curve,

$$\mathbf{r}(S_0, t) = (x(S_0, t), y(S_0, t)), \tag{2}$$

where $S_0$ is an arc length parameter for the (flat) mantle edge at the initiation of spine formation and $t$ is time. The shape evolves quasistatically in time, starting from a flat edge aligned with the $x$-axis, i.e. $\mathbf{r}(S_0, 0) = (S_0, 0)$, with $\mathbf{r}$ determined sequentially at times $t > 0$ by solving the equations of mechanical equilibrium, given in vector form by

$$\frac{\partial \mathbf{n}}{\partial S_0} + \mathbf{f} = \mathbf{0}$$

$$\frac{\partial \mathbf{m}}{\partial S_0} + \frac{\partial \mathbf{r}}{\partial S_0} \times \mathbf{n} = \mathbf{0}. \tag{3}$$

Here, $\mathbf{n}$ and $\mathbf{m}$ are the resultant force and moment in the rod, respectively, and $\mathbf{f}$ is an external force imposed by the current shell edge. The moment is related to the curvature $\mathbf{u}$ by a constitutive law. For a planar elastic rod, the moment and curvature have the form $\mathbf{m} = m\mathbf{e}_z$, $\mathbf{u} = u\mathbf{e}_z$, where $u(S_0, t)$ is the bending strain of the planar curve $(x(S_0, t), y(S_0, t))$, and $\mathbf{e}_z$ is a

unit vector in the direction orthogonal to the plane of the rod. We use the standard linear constitutive law, $m = E_b u$, where $E_b$ is the bending stiffness, equal to the product of the Young's modulus $E$ and the second moment of inertia.

Letting the shell edge be parameterised by $\mathbf{p}(S_0, t)$, the interaction between mantle and shell is modeled by $\mathbf{f} = k(\mathbf{p} - \mathbf{r})$, where $k$ describes the strength of the attachment, i.e., resistance to deformation of the mantle away from the shell edge. The calcification process of secreted shell material, and thus updating of the shell edge, is then modeled by the evolution equation:

$$\frac{\partial \mathbf{p}}{\partial t} = \eta(\mathbf{r} - \mathbf{p}), \tag{4}$$

where $\eta$ characterises the rate of calcification. The system is driven by growth of the mantle, i.e., a rule dictating the increase in the arc length. This is modeled within the framework of morphoelastic rods [34] by identifying a growth arc length parameter $S$, related to the initial arc length parameter, $S_0$, by the growth stretch, $\gamma$, via $\gamma = \partial S/\partial S_0$. Mantle growth is then described by an evolution equation for $\gamma$, taking the general form:

$$\gamma^{-1}\frac{\partial \gamma}{\partial t} = g(S_0, t, \ldots), \tag{5}$$

where the function $g$ describes the incremental increase in arc length, and may depend on position, time, and other factors. In particular, spatial inhomogeneity in growth is captured by $S_0$ dependence in $g$ and is a key feature of our analysis.

The presence of inhomogeneity in mantle growth/secretion and/or mechanical properties may be inferred from the highly localised appearance of spines in *T. sazae*, which is in contrast to the uniform wrinkling pattern that would be observed with a spatially homogeneous system. In fact, spatial inhomogeneity in spine production is likely present even within the spatial domain of a single spine: in the model for single spine formation presented by Chirat et al. [15], it was found that inhomogeneity was generally a necessary ingredient for generating realistic spine shapes. In their model, Chirat et al. [15] incorporated an inhomogeneity in the bending stiffness of the mantle, $E_b$, with a decreased bending stiffness at the spine tip needed for most spine shapes. Intuitively, since the curvature at the spine tip is highest, and much higher in the case of a sharp spine, such shapes can only naturally emerge, i.e. appear as minimising mechanical energy, if the bending stiffness is correspondingly lower.

In Chirat et al. [15], an inhomogeneous stiffness ($E_b = E_b(S_0)$) was combined with a uniform growth profile ($g(S_0) \equiv$ const in (5)), taken as decoupled inputs to the model. Here, we model the spine-formation process in the same basic way, but with several improvements. First, in the context of our multiscale framework, we include an inhomogeneous growth profile that is directly connected to the biochemical pattern obtained as the output of the system 1. Specifically, we suppose that the activator chemical $a(x, t)$ governs mantle growth, and thus the shape of the growth profile $g$ in (5) is taken to be proportional to the concentration $a$, scaled in an appropriate way (see S1 Text Section 3). Second, we directly link mantle bending stiffness to growth, as described in S1 Text Section 2, an assumption that produces realistic spine shapes in a way that emerges more naturally than in the model of [15]. With the growth profile governed by $a(x, t)$, the system is closed by imposing clamped boundary conditions at the ends of the rod, $S_0 = 0$ and $S_0 = L_0$, where $L_0$ is the initial length of the spine-forming region prior to growth.

More details on the tissue-level model are provided in S1 Text Section 2, including computational details on our numerical approach to solving the system of equations outlined in this section.

### 2.3 Best-fit spine parameter extraction

Having formulated a procedure to map from a point in the 6-dimensional input parameter space, $\mathcal{S}$, at the molecular level to the evolving shape of a spine at the tissue level, we require a procedure for connecting that shape to the profile of spines from actual shells and quantifying spine form. As noted, the general goal is to find the point in the space, $\mathcal{S}$, that produces a best-fit shape for a given shell.

However, it is important to note that the computational solving at the tissue scale is non-trivial, in particular because at each time step the equations of rod geometry and mechanical equilibrium form a nonlinear and inhomogeneous boundary value problem. Common techniques such as branch tracing do not apply, given the presence of the shell edge term, **p**, which acts as a non-uniform forcing term that must be updated quasi-statically between each solving step. For a given growth profile, $g$, and other input parameters, solving for the time evolution of **r** and **p**, i.e,. the evolving shape of the mantle and shell edge, takes on the order of minutes to tens of minutes. Therefore, when searching the 6-dimensional space $\mathcal{S}$ for the parameters that produce an evolving spine shape most similar to a given real spine, there is a computational bottleneck in solving the equations at the tissue scale.

**2.3.1 Workflow.** Fig 2 suggests a workflow that describes how the parameters, $\mathcal{S}$, produce a profile, $a$, that maps to a growth profile, $g$, that is then compared with a spine image, after which $\mathcal{S}$ is updated until an optimal shape agreement is obtained.

Our approach in searching the space $\mathcal{S}$ is to employ a Quality-Diversity algorithm [35, 36], a type of black-box optimization algorithm designed to produce a variety of candidate solutions, to generate the space of growth profiles generated by the molecular model (see Section 2.4 for details). Quality-Diversity algorithms have been established as an efficient way to explore a high-dimensional space and build a fitness landscape across phenotypic characteristics. Note that for low dimensional problems other methods such as Latin Hypercube Sampling or Sobol sequence can efficiently deal with this exploration phase, which are also provided by our framework.

Due to the tissue-level bottleneck, we restrict the class of functions that can be input at the tissue-level. In particular, we suppose that the growth profile, $g$, in Eq (5) has the shape of a Gaussian:

$$\gamma^{-1}\frac{\partial\gamma}{\partial t} = g(s) = g_1 \exp\left(\frac{-\left(s - \frac{l}{2}\right)^2}{2g_2^2}\right). \tag{6}$$

where $s$ is the arc length parameter in the current configuration, and $l$ the total current arc length. Since we restrict our analysis in this study to individual spines, a Gaussian provides an appropriate form to capture the heterogeneity needed for realistic spine formation. We note that a similar form was used in Chirat et al. [15] mechanical for the bending stiffness, $E_b$, but with an inverted Gaussian; since, in our model, the bending stiffness is inversely proportional to the cube of the growth, the same effect is achieved here. Moreover, as we show in Section 3, the two degrees of freedom $g_1$ and $g_2$, describing the growth rate and width of the growing region, respectively, provide sufficient variability to produce distinct but realistic spine morphologies. In this way, only two input parameters are needed at the tissue level. Other tissue-level parameters, such as the domain width, calcification rate, and adhesion stiffness, are set in non-dimensional form by a combination of estimating physical values and producing realistic buckling modes; for more details see S1 Text Section 2. Since our approach links the steady state spatial profile of the activator, $a(x, t)$, with the growth profile $g(s)$, we require that the biochemical model outputs profiles reasonably close to a single Gaussian. An advantage of the

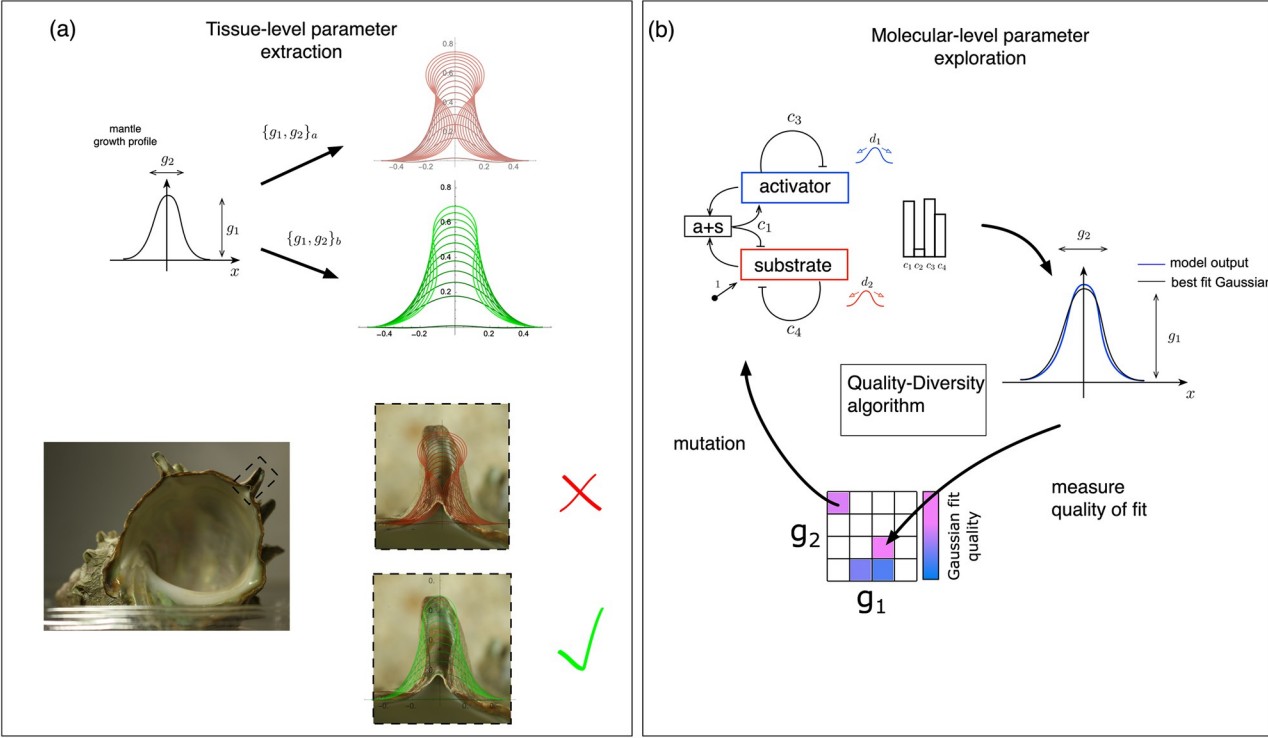

**Fig 3. Model workflow.** (a) At the tissue level, a Gaussian growth profile is input to the mechanical model, defined by two parameters $\{g_1, g_2\}$, for which different values produce different spine shapes that are then compared to a given spine image from a real shell in order to determine a best fit target pair $\{g_1, g_2\}$. (b) At the molecular level, a Quality-Diversity algorithm looks for the full range of $\{g_1, g_2\}$ pairs that can be produced by the Activator-Substrate reaction-diffusion system and stores the parameter sets that produced them in a grid. Sets in the grid may be reused by the algorithm, with some added noise (mutation), in an attempt to explore nearby area of the $\{g_1, g_2\}$ space. The filled grid is then used as a look-up table that provides parameter sets matching a given $\{g_1, g_2\}$ pair from the tissue level.

system (1) is that such profiles do emerge, and the requirement for a single Gaussian was used to constrain the biochemical parameter space (for more details, see S1 Text Section 3).

Rather than simulating the tissue model at each call within the optimization algorithm, we integrate before exploration the tissue-level model for different pairwise combinations of $g_1$ and $g_2$, thus producing a finite zoology of shape evolution curves that can be compared with real shells and providing a range of reasonable values for those parameters. This approach allows us to define both the quality and diversity for the exploration process: the quality of a given growth profile is defined as how close it matches a Gaussian shape and its diversity is defined as whether the corresponding $\{g_1, g_2\}$ pair of that Gaussian is different enough from solutions found so far. Note that $g_1$ and $g_2$ are continuous values and thus will likely not match exactly the values used in the initial integration. The final set of parameters kept by the algorithm (due to their quality, diversity, or both) is denoted by $\hat{\mathcal{S}} \subset \mathcal{S}$.

Once the exploration process is completed, we can extract from each spine image the values of $g_1$ and $g_2$ that best match the shape (details in Section 2.3.3). This provides a pair of target values of $\{g_1, g_2\}$ extracted for each spine; the parameter space $\hat{\mathcal{S}}$ is then searched with the objective of producing an output that is close to a Gaussian with width and height corresponding to the target $\{g_1, g_2\}$ values, allowing for an optimisation protocol that does not need to explicitly call the tissue model. This updated and simplified workflow is illustrated schematically in Fig 3.

**2.3.2 Shell collection and imaging.** We bought *T. sazae* from fishermen in Japan who dive in the vicinity of Jyogashima, Kanagawa prefecture, Najima in Hayama, Kanagawa prefecture and Okinoshima in Tateyama, Chiba prefecture, respectively. Hayama and Tateyama are relatively sheltered shores in the Sagami Bay and the Tokyo Bay, respectively, and suffering from a major coastal desertification in recent years [37]. On the other hand, Jyogashima is more exposed and has been recovering from a major coastal desertification since 2014 [38] and food sources for *T. sazae* are more abundant compared to Hayama and Tateyama.

Specimens were brought to the laboratory within the day when they are caught, and dissected to collect the shells. Photographs of the shells were taken using a digital camera (Canon EOS Kiss x7) with a micro lens (Canon EF100mm f/2.8L Micro IS USM) and magnified photographs of the spines were taken using dissection microscope (Nikon SMZ10) equipped with a digital camera (Canon EOS Kiss x7). We collected and photographed for analysis 12 specimens from Hayama, 16 from Jyogashima, and nine specimens from Tatayama.

**2.3.3 Parameter extraction.** In order to determine the best-fit values of $\{g_1, g_2\}$ for each spine, each magnified spine photograph was imported into Mathematica [39], where the shell profiles that were output from the tissue model were superimposed within a GUI environment (the `Manipulate` command in Mathematica). The model profiles were initially scaled, rotated and translated appropriately to match the size and orientation of the spine in the photograph, and then the values of $\{g_1, g_2\}$ were varied, while stepping through the time sequence of curves and comparing the profiles with the spine image. We note that the objective was not merely to match the final shape, but to best match the entire time sequence of spine growth. For most spines, growth lines from earlier portions of the spine were readily apparent; these provided profile curves that could be compared with model curves in the early and middle stages of spine production. In this way, we sought the values of $\{g_1, g_2\}$ for which the set of profile curves best matched the characteristics of the spine. Small spines, i.e. spines that had not attained much height at the point of capture, were rejected, as there was insufficient detail in their profile to accurately produce a match (almost any set of values could match reasonably well a spine that appears only as a small bump). To avoid confirmation bias, all shells were fit in sequence and prior to plotting any data. For more details on the fitting process, see S1 Text Section 5. In S1 Text Section 5, we also include a table of images of each spine that was fitted, with best-fit model curves overlaid as well as the corresponding values of $\{g_1, g_2\}$.

## 2.4 Algorithmic exploration of the parameter space

We use MAP-Elites [40], implemented in the Python library `qdpy` [41], to explore the parameter space. The algorithm generates sets of parameters for the model (see Table 1), which are then used to simulate the reaction-diffusion model (1). We fixed the diffusion coefficients as we do not expect to see variability on those parameters between individuals. Those parameters were instead set to generate appropriate growth profiles, *g*, when performing a cursory direct

**Table 1. Model parameters, parameter ranges, and sampling methods.**

| Parameter | Minimum value | Maximum value | sampling method |
|---|---|---|---|
| $c_1$ | $10^{-6}$ | 1 | log |
| $c_2$ | $10^{-3}$ | 2 | log |
| $c_3$ | $10^{-1}$ | 1.5 | linear |
| $c_4$ | $10^{-5}$ | $10^{-2}$ | log |
| $d_1$ | $10^{-2.75}$ | $10^{-2.75}$ | fixed |
| $d_2$ | $10^{-0.15}$ | $10^{-0.15}$ | fixed |

exploration of the parameter space. Parameter ranges and sampling methods are given in Table 1.

We then fit the spatial concentration profile of the activator species, $a(x, t)$, to a Gaussian function, giving us three values: the distance between the molecular profile and the fitted function and the parameters $g_1$ and $g_2$ from Eq 6. The set of parameters is then stored in a grid at a position defined by $g_1$ and $g_2$. If another set of parameters generated a similar Gaussian shape (i.e., near-identical values $g_1$ and $g_2$, based on the grid discretization), the algorithm only keeps the set generating the best fit. This way, we ensure that the generated parameter space $\hat{\mathcal{S}}$ only contains sets that produce concentration profiles that are close to a Gaussian profile. Once the algorithm has found enough sets, further sets may be generated through a small change in the parameters of a random existing set (mutation) or through complete random sampling (like the initial points). Both mutation and random generation sample the parameters either logarithmically or linearly, depending on their range and impact on the model. The exploration continues until the algorithm has evaluated a given number (budget) of parameter sets. Furthermore, to ensure the stability of parameter space $\hat{\mathcal{S}}$, the algorithm is rerun multiple times, generating independent grids. Parameters used for the algorithm and simulation are given in Table 2. Each grid is then used to map pairs $\{g_1, g_2\}$ to a given set of parameters for the biochemical model. Specifically, we look for the nearest non-empty cell in the grid and return the corresponding parameter set. Those sets are then analyzed to identify trends with respect to the different populations of shells.

## 3 Results

### 3.1 Impact of growth profile and spine shape

Before applying our analytical procedure to the shells obtained and investigating population-level differences, we first demonstrate the qualitative effects of the growth profile on the spine shape. Fig 4(a) shows a 'morphospace' of spine profiles for different values of $g_1$, $g_2$, where we recall that the incremental growth profile at the tissue level takes a Gaussian form

$$\gamma^{-1}\frac{\partial\gamma}{\partial t} = g(s) = g_1 \exp\left(\frac{-\left(s - \frac{l}{2}\right)^2}{2g_2^2}\right), \tag{7}$$

where $s$ denotes the current arc length parameter, which relates to the initial and grown arc length parameters, $S_0$ and $S$, respectively, by the total stretch, $\lambda$, via $\lambda = (\partial s/\partial S)(\partial S/\partial S_0)$; and $l$ is the total current length of the rod (see S1 Text Section 2). Included in Fig 4(a), at each point in the $\{g_1, g_2\}$ plane are the evolving spine profiles (green curves), as well as the corresponding growth profiles $g(S_0)$, plotted on the same scale for comparison (and centered on the axis). The function $g(S_0)$ describes the increase in arc length of the mantle edge as a function of initial position. A larger value of $g_1$ thus corresponds to a larger increase in arc length per time (for a given material section), while a larger value of $g_2$ corresponds to a wider growing region.

**Table 2. Algorithm and simulation parameters.**

| Parameter | Value |
|---|---|
| Runs | 5 |
| Budget | 50000 |
| Elite grid discretization | 100x100 |
| Batch size | 20 |
| Simulated time | 1000 time unit |

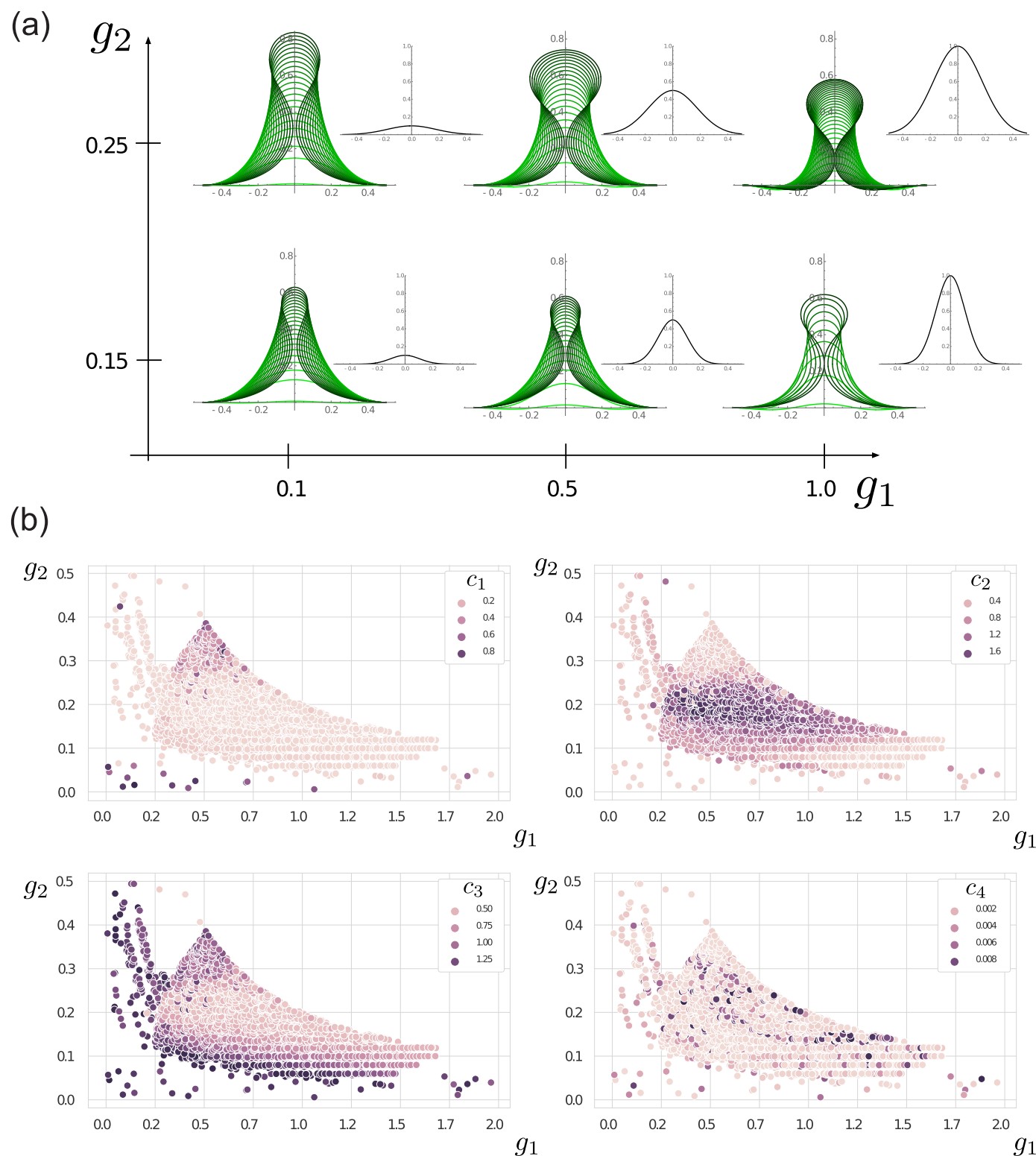

**Fig 4.** (a) Morphospace of spine shapes as the growth parameters $g_1$ and $g_2$ are varied. (b) Aggregated sets of biochemical model parameters, plotted with respect to their respective $\{g_1, g_2\}$ values. Coloring corresponds to the value of individual parameters, showing a clear global organization for $c_1$, $c_2$, and $c_3$.

Several features are evident in Fig 4(a). Increasing $g_1$ has the effect of producing a more curved spine, while a lower value produces a straighter and taller spine before self-contact occurs. This reflects the notion that with an increased excess of length (larger $g_1$), the mantle deformation per time step is more pronounced and therefore with higher curvature, while with smaller $g_1$, the mantle shape evolves more slowly, staying closer to the evolving shell edge. An increased $g_2$ produces a wider spine, since the growing region is wider, and in particular we find a much narrower spine tip for the smaller value of $g_2$, which is due to the combination of a narrower growing region, but also the corresponding decreased stiffness.

## 3.2 Impact of biochemical model on growth profile

Having established ranges of $\{g_1, g_2\}$ that generate realistic spine shapes, we consider next whether there are parameter regions in the biochemical model that output a concentration profile that can match the desired Gaussian shapes.

Fig 4(b) shows an aggregation of all sets found through the exploration of the biochemical model's parameter space. Contrastingly, white space correspond to $\{g_1, g_2\}$ values that could not be generated by any of the parameters. We observe that the level of parameters, $c_1$, $c_2$, and $c_3$, has a strong connection to the $\{g_1, g_2\}$ values, forming distinct regions. That is particularly true with respect to the values of $g_2$. Low $c_1$ (low saturation of the autocatalytic production) combined with low $c_2$ (low direct production from substrate) and high $c_3$ (high degradation rate) yields low values of $g_2$. When $c_1$ is high instead (strong saturation of the autocatalytic production), we get high $g_2$ values. Finally, intermediate values of $g_2$ are achieved for high $c_2$ values with low values of $c_1$ and $c_3$, thus hinting at a strong activation, both from autocatalysis and production from the substrate.

The parameter, $c_4$, however, seems less well-defined and can seemingly take a wide range of values regardless of $g_1$ or $g_2$. This connection can be explained by considering that, in the biochemical model (1), $c_1$, $c_2$, and $c_3$ are connected through their common impact on the concentration of the activator species, $a$. However, $c_4$ only impacts the substrate species, $s$, possibly allowing more freedom compared to the other parameters.

Finally, we note that the sets tend to gather in a central area with well-defined borders, with only a few points extending beyond. The sparsity of these areas can be attributed to two possibilities: (1) the molecular model is unstable for such parameter sets or (2) such $\{g_1, g_2\}$ values cannot be generated by the model at all. The ruggedness of the main cluster's bottom edge and the presence of a few points for low values of $g_1$ and $g_2$ tend to point towards possibility (1) in that area. Adding a bias to the exploration algorithm, as well as finer parameter tuning for those area, could help flesh out the parameter space. On the other hand, the sharpness of the upper right edge seems to indicate the latter possibility; that indeed, high values of both $g_1$ and $g_2$ cannot be generated by the model. That hypothesis is consistent with a standard linear stability analysis of the system (see S1 Text Section 4).

## 3.3 Identifying and quantifying morphological differences

The main results of our analysis are summarised in Fig 5. Fig 5(a) shows the result of the extracted $\{g_1, g_2\}$ values from spine images, with the radii of the circles scaled by the number of spines with those particular best-fit values of $\{g_1, g_2\}$. Fig 5(b) plots the corresponding values of $c_i$, $i = 1 \ldots 4$ that map to the given $\{g_1, g_2\}$ in relation to each other. In each case, the values are sorted by population: orange points correspond to Hayama, blue points correspond to Jyoga-shima, and green points correspond to Tateyama. Here we note that these plots aggregate the values from all 5 runs of the optimization algorithm. While few $\{g_1, g_2\}$ pairs were observed for each population, the four $c_i$ parameters offer enough flexibility to generate different values for

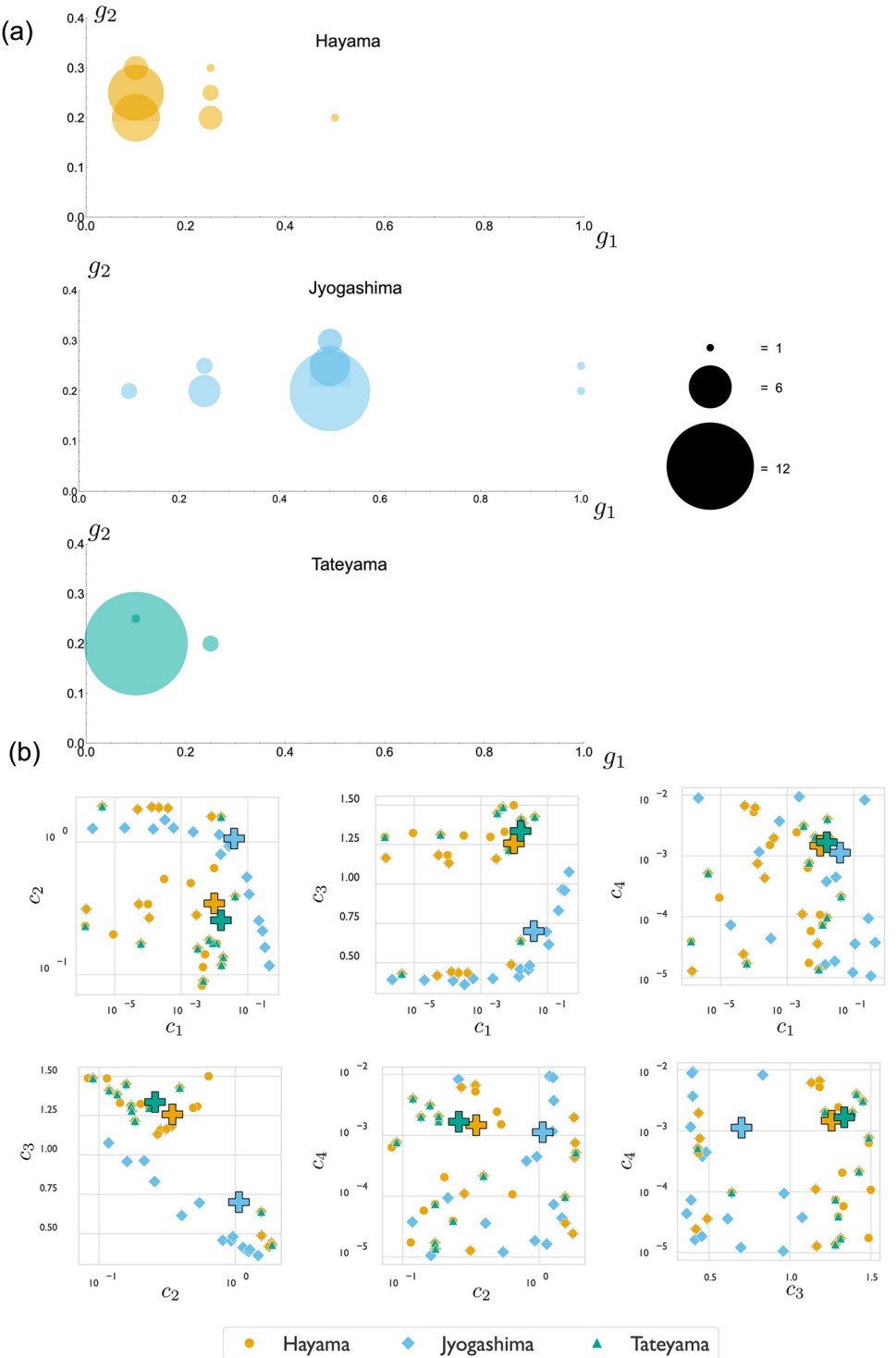

**Fig 5.** (a) The best-fit values of $\{g_1, g_2\}$, extracted from comparing tissue-model output to images of real shells, separated by population. (b) Relationship between biochemical model parameters for each population, as obtained from the algorithmic parameter exploration. The + symbol represents the average parameter values for the population, highlighting the difference between the Jyogashima population and the other two.

each independent run. However, as noted in the previous section, a general trend could still be observed in the relation of the parameters with each other.

In Fig 5(a) we find a population-level bias in $g_1$ values: spines on shells collected in Hayama and Tateyama are better modeled by smaller values of $g_1$, while spines from shells collected in Jyogashima are better modeled by larger values of $g_1$, though these shells also show a wider distribution. Using two-sided Wilcoxon rank-sum tests, the distribution of $g_1$ values are, on average, an estimated value of 0.25 higher compared to Hayama, with a p-value of $1.202 \times 10^{-5}$ (and 0.4 higher compared to Tateyama, with a p-value of $5.7 \times 10^{-6}$). The $g_1$ values for Hayama and Tateyama do not show a significant statistical difference, and similar analysis of the distributions of $g_2$ values show only very small statistical difference between the populations (see S1 Text Section 6 for details).

This implies that the only statistically significant difference is found between the distributions of $g_1$ values, with the distribution of Jyogashima values being larger than the other two populations. Recall from Fig 4 that $g_1$, which controls the rate of growth/secretion, primarily impacts the height and curvature of the spine profile, with larger values of $g_1$ producing a more curved spine that reaches the point of self-contact at a lower height. Note that this does not imply that spines with larger $g_1$ are shorter, as our computational model stops at the point of self-contact, while spine formation may continue well past self-contact—that is, in some cases, the two sides of the spine come into contact, beyond which a sort of 'zippering up' occurs while the spine continues to grow in height (see also S1 Text Section 5). This analysis thus suggests that a difference in spine form does exist between the shells we have collected from Jyogashima and the shells collected from the other two populations. While shells from the populations generally appear quite similar, upon closer inspection, it can be observed that the spines of shells we have collected from Jyogashima tend to be more curved, while the shells from Hayama and Tateyama tend to have straighter spines. The distinction, though subtle, can be observed in the spines of the sample shells shown in Fig 1 (see also S1 Text Section 5, which shows images of all spines for which we have extracted data, as well as the best-fit curves overlaid).

The population-level differences in $g_1$ between the Hayama/Tateyama populations and the Jyogashima population is reflected in the biochemical parameters $\{c_1, c_2, c_3, c_4\}$. This is most evident in examining the average parameter values for the populations, indicated by the + sign in the plots of Fig 5(b). We see that in all parameter pairings, the averages for Hayama and Tateyama are closer together and separated from the average of Jyogashima. The difference is greatest in the parameters $c_2$ and $c_3$: the Jyogashima population shows a higher value of $c_2$ and lower value of $c_3$. This is consistent with the biochemical interpretation of these parameters. The parameter $c_2$ describes direct production rate of the activator from substrate, while $c_3$ describes activator degradation. Hence high $c_2$ and low $c_3$ combine to make an increased concentration of activator, i.e. high $g_1$, and the high growth rate produces a more curved spine shape. No such trend could be observed in the values of $c_4$, with all populations having a similar average. Stability analysis showed that, as long as $c_4$ is low enough, its impact is minimal on the production of a Gaussian curve. When $c_4$ is low enough, the concentration of substrate remains high enough to help produce the activator, which is required to form a pattern.

## 4 Discussion

Connecting genotype to phenotype is a monumental task that crosses vast spatial and temporal scales, from gene expression to cellular activity, to tissue-level mechanics, to organ/organism-level form and function, to population level fitness pressures. This task is made even harder when dealing with non-model organisms, for which almost no known genetic networks and

protein-protein interaction networks have been established. Constructing analytical tools that may shed light on developmental pathways in such organisms will be of great benefit, and multiscale computational modeling has an integral role.

In this paper we have devised a set of models and a framework of analysis to investigate spine shapes in the mollusc species *T. sazae*. By linking a tissue-level mechanical model with a two-species biochemical model, we were able to uncover a statistically significant difference in tissue-level parameters between shells from one location (Jyogashima) and the other two from which we collected shells (Hayama and Tateyama). The difference could be traced to parametric differences in the underpinning biochemical model, providing some insight as to how the difference in form may have manifested.

The natural next step is to link the differences in the biochemical model to environmental differences between the locations. However, this is not possible at present, in particular due to the phenomenological basis of our biochemical model, which lacks a detailed biological foundation. Nevertheless, our results tell a consistent story, from which we can at least speculate on the environmental link. Computationally, the primary finding from our analysis was that shells in Jyogashima had a larger value of the growth rate, $g_1$, than Hayama and Tateyama, which corresponded most strongly at the molecular level to larger values of the substrate production rate, $c_2$, and lower values of activator degradation rate, $c_3$. These parametric differences effectively mean that specimens in Jyogashima were growing the spine more quickly, due to some combination of increased production rates or decreased degradation rates of the molecules responsible for shell production. This may be due to availability of food, since we observe recovery from coastal desertification [38] and more seaweed in the Jyogashima area than other localities, where coastal desertification has become a serious problem [37]. Though also of interest is that specimens in Hayama and Tateyama are exposed to calmer waters compared to Jyogashima. It remains an interesting direction for future research to investigate exactly which environmental differences play the strongest role in spine form, and how these differences are mechanistically translated into developmental processes.

As noted in the introduction, not all specimens grow spines at all, and as has been previously proposed [22], a change in environment could have turned off the spine-growing machinery. While our analysis has focused specifically on shells that produce spines, it is worth noting that many parameters will not produce a biochemical pattern at all, which we may reasonably correlate with no spine production. The biochemical parameters that we have fit to the spine shapes are generally very close to the boundary beyond which no pattern is produced, so again we see a consistent story: a small change in biochemical patterns, due to some underlying change in environment, can easily transition the system into a state for which pattern formation halts.

While our analysis and results tell a consistent story, we must also emphasize the significant simplifying assumptions made in our models. As well as employing a biochemical model that was chosen for phenomenological reasons with no specific biological justification, we have also posited the simplest possible link between biochemistry and tissue growth. Moreover, while shell secretion occurs simultaneously along the entire shell aperture, and with spines appearing periodically in time, we have focused our analysis on a single isolated spine. For these reasons, we must be careful not to draw any concrete biological conclusions on chemical or mechanical mechanisms underlying observed differences in spine form. Nevertheless, our study provides a proof of concept, upon which our group plans to extend/build in the future, in several ways. For one, we may extend the tools we have developed to explore spine formation in a 'whole shell' context, incorporating spatio-temporal modeling that explicitly accounts for both the timing and structure of multiple spines as they appear through development. Another important direction is to connect more directly to cell biology. One possible avenue is

to conduct generate and analyze transcriptome data of mantle cells, using either bulk or single-cell RNA-sequencing [42, 43], thereby generating initial data from which candidate gene regulatory networks driving spine formation can be inferred, ultimately informing more realistic molecular modeling.

Finally, we note that while the framework we have presented is motivated by and specifically constructed around a specific phenotype in a specific organism, the main ingredients in our investigation—a variation in form, emerging due to a combination of heterogeneous mechanical forces during growth and an underlying biochemical signal—are ubiquitous. In principal, the techniques developed here may provide, at least conceptually, a blueprint for analyzing morphogenetic variation between populations in other systems, e.g. leaf structure [44], horns and teeth [45, 46], or anatomical features such as nose and ears [47, 48], to name a few. Here, it is important to emphasize the 'plug and play' aspect of our model, i.e. different models at the tissue or biochemical level could be inserted, and the algorithmic steps of connecting the scales would work the same.

## Supporting information

**S1 Text.** This supplementary file contains a detailed description of the biochemical model; further details on the mechanical model, including computational details; a detailed description of linking the biochemical and mechanical models; a computation of the stability boundary for the biochemical model; extracted parameter values for spines, including spine images with best fit mechanical curves overlaid; a description of statistic testing of the spine parameters; and detail on the mathematical construction of full shell simulation. (PDF)

## Author Contributions

**Conceptualization:** Derek E. Moulton, Nathanaël Aubert-Kato, Atsuko Sato.

**Data curation:** Derek E. Moulton, Nathanaël Aubert-Kato, Axel A. Almet, Atsuko Sato.

**Formal analysis:** Derek E. Moulton, Nathanaël Aubert-Kato, Axel A. Almet.

**Investigation:** Axel A. Almet.

**Methodology:** Derek E. Moulton, Nathanaël Aubert-Kato, Atsuko Sato.

**Writing – original draft:** Derek E. Moulton, Nathanaël Aubert-Kato, Axel A. Almet, Atsuko Sato.

**Writing – review & editing:** Derek E. Moulton, Nathanaël Aubert-Kato, Axel A. Almet, Atsuko Sato.

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
