## [Decision Letter · Decision Letter 0]

2 Aug 2023

Dear Prof Moulton,

Thank you very much for submitting your manuscript "A multiscale computational framework for the development of spines in molluscan shells" for consideration at PLOS Computational Biology.

As with all papers reviewed by the journal, your manuscript was reviewed by members of the editorial board and by several independent reviewers. In light of the reviews (below this email), we would like to invite the resubmission of a significantly-revised version that takes into account the reviewers' comments. Please note that the critiques are substantial and require significant revision, and that the opportunity to revise is not a guarantee of ultimate acceptance.

We cannot make any decision about publication until we have seen the revised manuscript and your response to the reviewers' comments. Your revised manuscript is also likely to be sent to reviewers for further evaluation.

Sincerely,

Kirsten Hendrika ten Tusscher, PhD

Academic Editor

PLOS Computational Biology

James O'Dwyer

Section Editor

PLOS Computational Biology

Reviewer's Responses to Questions

**Comments to the Authors:**

Reviewer #1: This paper presents a new framework for connecting two levels of biological modelization: the molecular and tissue levels. The framework is applied to molluscan shell growth, specifically focusing on spine formation. At the molecular level, an Activator-Substrate model is used, where all molecules are aggregated into two elements: an actuator and a substrate. A reaction-diffusion mechanism is implemented, incorporating diffusion constants in non-dimensional equations. The tissue level represents the resulting spine growth through a parameterized curve. The two levels are connected through the activator equation, which regulates mantle growth within the parameterized curve. The framework aims to generate a "finite zoology of shape evolution curves" by varying two parameters of the parameterized curve. The best matches with biological data are selected to explore potential molecular dynamics that could produce corresponding curve parameters. This exploration is carried out using the MAP-Elite algorithm, designed for exploring large search spaces.

The results first demonstrate the morphospace achievable with this framework and conduct a parameter study to obtain various shapes. Subsequently, the molecular scale is explored by evaluating the necessary conditions for obtaining specific shapes observed in biological samples. The study identifies some necessary conditions for producing the diversity of shapes found in different environments where mollusks are found. However, some of these conditions propose hypotheses that are challenging to validate with real-world data and experiments.

The proposed framework is of interest to the scientific community as it represents a rare example of a multi-scale model for a biological process. Both models are simplistic to demonstrate the feasibility of the approach. However, the paper should clarify the biological plausibility of the obtained results, which is discussed in the conclusion. It would be beneficial to introduce this aspect more explicitly from the beginning. Additionally, the rationale for selecting mollusk shell growth as the specific use-case should be better explained.

Furthermore, the paper should provide more precision regarding several choices made in the framework. For instance, the interconnection between the two models (lines 284-286) lacks justification and clarity. The choice of the chemical used and the impact of proportional concentration on the activator equation should be explored in detail, as it can significantly affect the observed behavior and serves as a crucial element in the multi-scale connection.

Another aspect requiring clarification is the pre-integration of the tissue-level model for different pairwise combinations of parameters g1 and g2 (lines 341-342). While this approach reduces computational costs, it raises questions about the multi-scale nature of the study. It is unclear whether the molecular level can produce {g1, g2} values that were not generated, potentially resulting in different shapes than those used for evaluation from the zoo.

The use of the MAP-Elites algorithm for exploring parameter sets of the curve is not justified in the paper. It would be helpful to explain why this algorithm was chosen over other approaches such as parameter sampling (e.g., Latin Hypercube Sampling) and provide a rationale for its suitability in this context.

In section 2.3 (lines 289-290), the paper mentions fixing the diffusion coefficients due to the expectation of no variability between individuals. However, the impact of this choice is not addressed. Conducting a sensitivity study on these parameters would help justify such a simplification.

Furthermore, in section 3.1, the impact of growth profile and spine shape is proposed, yielding interesting results. However, the methodology lacks details regarding how the values of the pair {g1, g2} were explored. Providing more information on this aspect is necessary to assess the quality of the obtained results.

Lastly, SM Section 5 lacks convincing fitting to biological data. While some of the curves match the observed photos, others deviate significantly (e.g., Figure 2: spines 2A, 4Aii, 5B, 10A; Figure 3: spines 6A, 7A, 15B; Figure 4: spines 8A and 9A). Since the spine fitting is done manually, it raises questions about the impact on the final results of the proposed methodology. It should be explicitly stated whether different experts would produce spines with similar values.

In conclusion, while the framework presented in the paper is of interest to the scientific community, it lacks sufficient details to fully assess the quality of the methods. Therefore, I recommend a major revision before publication.

Reviewer #2: Summary:

The authors suggested a new computational framework to predict how biochemical conditions might differ for spines of Turbo sazae that live in different locations in Japan (Hayama, Jyogashima and Tateyama). First, they evaluate morphoelastic rods' shape by changing growth parameters, g1 and g2. Second, the growth parameters of the actual spines are predicted by comparing the actual shape with the generated rod shape. Third, they used the biochemical model based on the reaction-diffusion equations to see which biochemical parameters would result in the same profile as the growth parameters. Their results showed that spine shape can be divided into two groups (Hayama/Tateyama and Jyogashima). Based on their framework, they concluded that different biochemical parameters might be the cause of different spine shapes.

Although I agree that it is important to understand how biochemical conditions impact the animal's shape during development, I do not think that the current framework can be used to explain the morphological differences since there are too many assumptions, as I pointed out in the major comments. The manuscript was not friendly to readers who are not in the field, and it was also difficult to read due to the errors listed in the minor comments. The following points need to be addressed for acceptance.

Major Comments:

1. I am concerned about the usefulness of the current approach since the approach relies on many assumptions. There are many assumptions necessary before starting the optimization, and we need to select the following materials:

- reaction-diffusion equation: Gray-Scott model, activator-substrate model...

- cause of buckling: non-uniform growth, non-uniform mantle stiffness...

- what makes the spine shape different: genotype, phenotype, local environment...

- biochemical-mechanical coupling: an infinite number of combinations

The current conclusion can be drawn since we initially assumed that the dynamics are governed by the biochemical-mechanical coupling introduced in this paper. Since different sets of assumptions and equations would lead to different conclusions, I do not believe this approach can be used to understand the complex mechanism of shape formation. For example, there is a possibility that the spines of the two groups differ simply due to the local mechanical environment. I understood that the current framework can be used only when the assumptions are confirmed, which would be difficult to be achieved. I would like the authors to answer how they treat this problem in the manuscript.

2. As far as I understood, the mechanical model introduced on page 9 was not used, and the morphoelastic rod is utilized to simulate the shape evolution. Why is the model on page 9 introduced?

3. The manuscript was not friendly to readers not in this field since there were not enough explanations on the "secretion process" and "mantle" with schematics. There should be explanations of the technical terms with a simple schematic.

Minor Comments:

1. Line 128: no link to Reference

2. First paragraph of section 2.1.2: This paragraph is difficult to understand because the "biological facts" and "assumptions for the simulation" are explained at the same time. It would be helpful to clearly state the facts first and explain the assumptions afterward.

3. Line 249: z-axis was not defined

4. Line 294: "to solving" to "to solve"

5. Line 311: "there is a huge bottleneck at the tissue scale" does not make sense. The computational time of the tissue numerics is the bottleneck.

6. Eq. (6): s and l are not defined

7. Line 340: "Rather than simulate" to "Rather than simulating"

8. Line 341: define what is "pre-integrate"

9. Line 348: \\hat{S} is not defined

10. Line 462: I could not understand the sentence: "The ruggedness of the main cluster's bottom edge and the presence of a few points for low values of g1 and g2 tend to point towards the former possibility in that area". What is the former possibility in that area?

11. Line 463: numbers of g1 and g2 are not subscripts

12. Line 480: previous Section

13. Line 497: Why is the sentence "Note that this does not..." parenthesized?

14. Line 499: What do you mean by "continues well past self-contact in a number of spines"?

15. Line 566: missing a period

16. Figures in Appendix: Figure titles are only numbers

17. Equation number in Appendix: It should be like (A1) to avoid overlap with the main text

18. Is the quality-diversity algorithm used only for molecular-level? or, also for tissue-level?

**Have the authors made all data and (if applicable) computational code underlying the findings in their manuscript fully available?**

Reviewer #1: Yes

Reviewer #2: Yes

PLOS authors have the option to publish the peer review history of their article (what does this mean?). If published, this will include your full peer review and any attached files.

Reviewer #1: **Yes: **Sylvain Cussat-Blanc

Reviewer #2: No
---

## [Decision Letter · Decision Letter 1]

16 Jan 2024

Dear Prof Moulton,

We are pleased to inform you that your manuscript 'A multiscale computational framework for the development of spines in molluscan shells' has been provisionally accepted for publication in PLOS Computational Biology. 

Best regards,

Kirsten Hendrika ten Tusscher, PhD

Academic Editor

PLOS Computational Biology

James O'Dwyer

Section Editor

PLOS Computational Biology

Reviewer's Responses to Questions

**Comments to the Authors:**

Reviewer #2: The authors have addressed all the points I raised, and I support publication. For instance, the new Supplementary Figure 1 supports the reliability of this framework and is a good addition.

I would like to add a note regarding my initial comment. From the first round of reviewing, I was supporting the importance of proposing a new framework and found it original and interesting. However, I was raising a question about the value of delving deep into the analysis based on many assumptions (Fig 5). The toned-down claims in the revised version and the authors' reply have since addressed these concerns satisfactorily.

**Have the authors made all data and (if applicable) computational code underlying the findings in their manuscript fully available?**

Reviewer #2: Yes

PLOS authors have the option to publish the peer review history of their article (what does this mean?). If published, this will include your full peer review and any attached files.

Reviewer #2: No

---

## [Editor Report · Acceptance letter]

12 Feb 2024

PCOMPBIOL-D-23-00619R1 

A multiscale computational framework for the development of spines in molluscan shells

Dear Dr Moulton,

I am pleased to inform you that your manuscript has been formally accepted for publication in PLOS Computational Biology. Your manuscript is now with our production department and you will be notified of the publication date in due course.

With kind regards,

Anita Estes
